# Development and Application of a Remote Monitoring System for Agricultural Machinery Operation in Conservation Tillage

Changhai Luo [1,†], Jingping Chen [1,†], Shuxia Guo [1], Xiaofei An [1], Yanxin Yin [1], Changkai Wen [1], Huaiyu Liu [1], Zhijun Meng [1,*] and Chunjiang Zhao [2,*]

[1] Intelligent Equipment Research Center, Beijing Academy of Agriculture and Forestry Sciences, Beijing 100097, China
[2] National Engineering Research Center for Information Technology in Agriculture, Beijing 100097, China
* Correspondence: mengzj@nercita.org.cn (Z.M.); zhaocj@nercita.org.cn (C.Z.)
† These authors contributed equally to this work.

**Abstract:** There is an increasing demand for remote monitoring and management of agricultural machinery operation in conservation tillage. Considering the problems of large errors in detecting operation quality parameters, such as tillage depth and corn straw cover rate, in complex farmland environments, this paper proposes a tillage depth measurement method based on the dual attitude compound of a tractor body and three-point hitch mechanism with lower pull rod and an online measurement method based on K-means clustering of the corn straw cover rate on farmland surface. An operation monitoring terminal was developed for the remote collection of quality parameters of conservation tillage field operation. A remote monitoring system of agricultural machinery operation was constructed and applied over a large area. The field tests showed that the static mean error and root-mean-square error of this method were 0.16 and 0.67 cm for uphill and 0.36 and 0.57 cm for downhill, respectively. For the 28 and 33 cm tillage depth tests, the mean dynamic measurement errors of this method were 0.55 and 0.61 cm, and the root means square errors were 0.64 and 0.73 cm, respectively, and the coefficient of variation of tillage depth did not exceed 3%. The correlation coefficient between the corn straw cover rate detection algorithm based on K-means clustering and the manual image marking method reached 0.92, with an average error of 9.69%, and the accuracy filled the demand for straw cover rate detection. The detection accuracy of tillage depth and straw cover rate was high and thus provides an effective means of information technology support for the quality monitoring and production management of conservation tillage farming operations.

**Keywords:** conservation tillage; subsoiling operation; operation quality; tillage depth; corn straw cover rate; operation remote monitoring

## 1. Introduction

Conservation tillage is one of the leading technologies for sustainable agricultural development, mainly through the implementation of no-tillage and straw mulching to control soil, wind, and water erosion and dust pollution and improve soil fertility and soil moisture storage capacity, and it has been promoted and developed in many countries [1–3]. Subsoiling operation, straw return, and no-till precision sowing are critical support technologies of conservation tillage. They are directly related to the quality of agricultural production operations, operational effectiveness, and operational management efficiency [4,5]. Therefore, the online monitoring and remote supervision of conservation tillage-related working conditions for the operation area are significant advancements to improve the efficiency of agricultural production. Based on advanced measurement and control technology, system analysis methods, and big data intelligent processing theory, it is essential to realize online sensing and remote monitoring of operation quality of critical links, such as deep pine and straw returning to fields [6,7].

Subsoiling operation and straw return are core components of conservation tillage. Accurate online detection of subsoiling operation tillage depth, intelligent detection of corn straw cover rate, and remote monitoring can improve the effectiveness of conservation tillage. The online detection of tillage depth has become a research hotspot. Yin proposed a tillage depth detection system based on the tractor three-point hitch mechanism with lower tie rod attitude measurement, which realized the real-time measurement of tillage depth of flat slope and micro-slope farmland [8]. However, the error in measuring tillage depth on the ground with a slope greater than 6° was significant. In a later study, Yin combined attitude sensors to collect the horizontal attitude data of the tractor's lower pull rod and subsoiler [9]. This method could achieve real-time measurement of the tractor operation plowing depth of steeper slope farmland, with an average error of plowing depth detection of 0.45 cm. However, the calibration process for this method is complicated, the error compensation is difficult to determine, and the operation and maintenance requirements of the system are high. In addition, it is challenging to realize the quality online supervision for the subsoiling operation of many tractors based on the data platform, which is not conducive to promoting large area applications. Based on the sensor fusion approach, Kim et al. developed a tillage depth monitoring system consisting of a linear potentiometer (CLS1322, ActiveSensors, Christchurch, UK), inclinometer (SST141, Vigor Technology, Shanghai, China), and optical distance sensor (ODSL 9/C6-650-S12, Leuze electronic, Owen, Germany). However, the accuracy and stability of the tillage depth detection system in field trials have not been verified [10]. Bentaher et al. measured three orthogonal components of tillage force by three force sensors of the tractor suspension system to improve the tillage depth monitoring system [11]. However, the tillage depth value obtained by this method is relative, and changes in the structure of the farm implement can lead to changes in the conversion relationship. As shown above, most existing tillage depth monitoring methods and systems are based on the information of the three-point hitch mechanism and do not consider the influence of the tractor body on the tillage depth of the hooked implements or the influence of terrain with a steeper slope on the tillage depth measurement.

Straw is an important by-product of agricultural production. Straw returning to the field can prevent and control dust, water and soil loss, water storage and soil moisture conservation, and soil fertility. The corn straw coverage rate is an essential technical indicator for the effect of straw returning to the field and is a basis for the subsidy of straw returning to the field [12]. However, the corn straw coverage rate is mainly measured by artificial methods, such as the rope pulling method. The measurement data of the corn straw coverage rate are greatly affected by human subjective factors and have low efficiency.

Some scholars have used satellite remote sensing to select the short infrared band sensitive to straw cellulose and lignin to establish a large-scale monitoring system for straw coverage based on remote sensing [13,14]. However, this telemetry method monitors only a single parameter. It cannot achieve simultaneous monitoring of operational status, operational surface, and other quality parameters with the rest of conservation tillage, which limits the integration of remote sensing monitoring of straw cover and field agricultural machinery operation monitoring systems [15]. In addition, scholars have conducted studies using image processing methods [16]. Li et al. combined the entropy value of texture features with a back propagation neural network for straw cover detection [17,18]. Jafari et al. converted the images into different color fields for straw identification and achieved an error of 2.3%; however, the method was susceptible to interference from the camera angle and camera environment [19]. The above methods process straw return images offline without real-time monitoring [20–22]. In contrast, Liu et al. established a real-time monitoring method for corn straw cover rate based on image recognition of field operations, effectively supporting online monitoring of corn straw cover rate during straw return operations. However, a complete operational processing method from data collection, back transmission, storage analysis, quality evaluation, to specific physical systems has yet to be developed [23–27].

This paper proposes the methods of subsoiling operation tillage depth measurement and farmland surface corn straw cover rate detection under a complex farmland environment with the aim of meeting the operation quality and information management requirements of conservation tillage agricultural machinery widely used in Northeast China. We developed an operation monitoring terminal suitable for remote collection of quality parameters of conservation tillage field operation, constructed an operation monitoring and management system, and carried out the large-scale application for conservation tillage. The tillage depth measurement method effectively integrated the information of the tractor body and the lower pull rod of the three-point hitch mechanism, and the straw cover rate detection method took the K-means clustering theory as the core.

## 2. Overall Scheme of a Remote Monitoring System for Agricultural Machinery Operation in Conservation Tillage

The remote monitoring system for agricultural machinery operation in conservation tillage mainly consisted of a tillage depth detection device, a corn straw coverage rate detection unit, an operation monitoring terminal, and a service system for supervision and management of remote operation. The tillage depth detection device consisted of two attitude sensors and an agricultural equipment identification sensor. Among them, to collect attitude information of the tractor body and lower pull rod, two attitude sensors were installed, one in the tractor cab and the other in the lower pull rod. In addition, the basic structural parameters of the subsoiler were stored in the equipment identification sensor (AgChip Science and Technology (Beijing) Co., Ltd., Beijing, China). Finally, the information from the three sensors was integrated to obtain the status and tillage depth of the subsoiling operation. The straw cover detection and collection unit consisted of two groups of cameras. These groups were installed in the front counterweight block of the tractor and the back of the tractor cab roof, respectively. The two groups of cameras were connected to the monitoring terminal to obtain the image data of the farmland surface in front and behind the tractor operation direction. An operation online monitoring terminal was installed in the tractor cab. This terminal can analyze and integrate information, such as tillage depth data, image data, location information, operation speed, and operation status. The processing results can be displayed on the monitor terminal and transmitted to the service system for remote operation supervision and management in real-time through a 4G wireless network. The system can send, receive, parse, and store operation data from the monitoring terminal. The system can calculate the operation area and corn straw coverage rate that have reached the standard. In addition, the system can evaluate the quality of tillage operations and the amount of corn straw coverage rate for no-till seeding on the operating plots. The overall scheme of the system is shown in Figure 1.

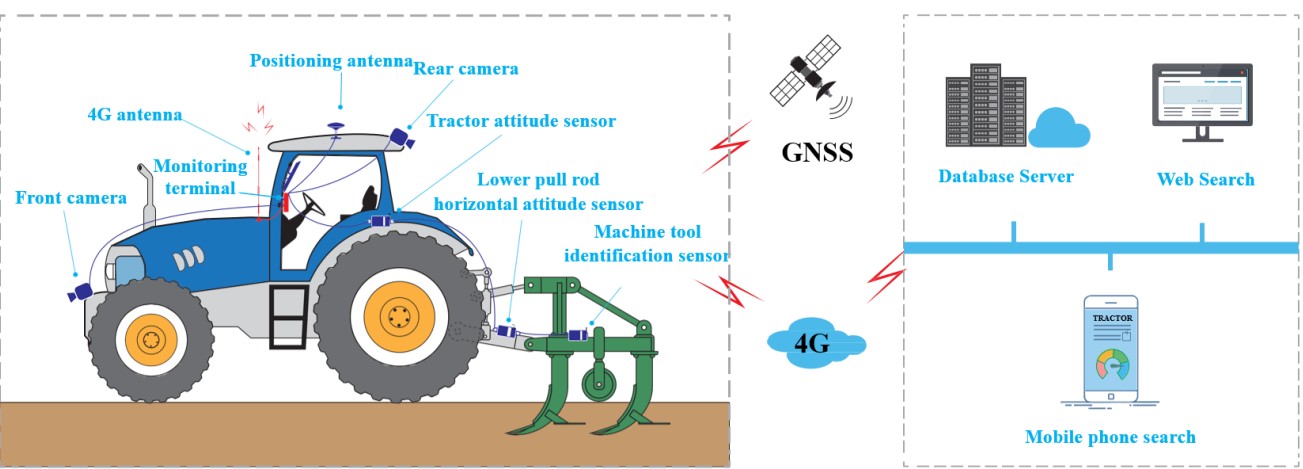

Online monitoring terminal for conservation tillage      Remote operation monitoring service system

**Figure 1.** Overall plan of the system.

### 3. Methods for Testing the Quality of Agricultural Machinery Operations in Conservation Tillage

*3.1. Measuring Tillage Depth in Complex Terrain*

　　　Considering that tillage depth detection in slope and complex environments is easily affected by attitude change and vibration and that accuracy is difficult to guarantee, a tillage depth measurement method based on the body and lower pull rod of the three-point hitch mechanism was proposed. This method integrates dual attitude fusion and sensor data processing to solve the problem of slope tillage depth detection. Based on the dual attitude sensor (AgChip Science and Technology (Beijing) Co., Ltd., Beijing, China), this paper monitored the change of the attitude of the tractor body and its lower pull rod in real time. The geometry of the three-point hitch mechanism was combined to build a tillage depth monitoring model for subsoiling operation. Thus, the influence of the slope angle on tillage depth detection is reduced by the horizontal attitude of the tractor body to detect subsoiler operation depth under complex terrain.

　　　Suspended subsoilers are used for field tillage under the traction of the tractor's three-point hitch mechanism. The front end of the hitch mechanism's lower pull rod and the tractor's lower hitch point are hinged through a connecting unit, and the center link point is A. The other end is hinged to the subsoiler through a connecting unit, and the center point of the hinge is B. The front end of the upper pull rod is hinged with the upper hitch point of the tractor (China YTO Group Co., Ltd., Luoyang, China), and another end is hinged with the upper hitch point of the subsoiler. The rigid suspension unit holds the equipment rigidly to the tractor. The tractor controls the lifting arm through the hitch mechanism to adjust the working depth of the subsoiler. The tillage depth detection principle of the suspended subsoiler is shown in Figure 2.

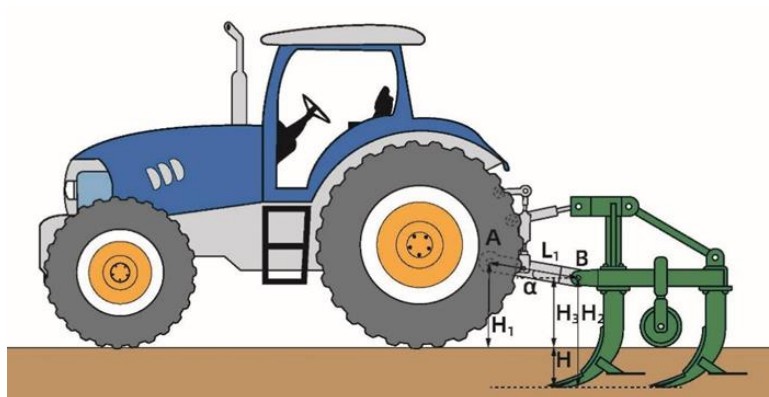

**Figure 2.** Detection principle of suspended subsoiler tillage depth.

　　　When a tractor unit is placed on the open and flat ground, the vertical distance between the lower link point A and the horizontal plane is $H_1$, the vertical distance between the lower hitch point B and the horizontal plane is $H_2$, and the straight-line distance between the points A and B is the length $L_1$ of the tractor lower pull rod. When the tractor unit is operated on a tillage operation with a slope lower than 2°, as shown in Figure 2, the tractor unit's operation depth $H$ is calculated using Formula (1):

$$H = H_2 - (H_1 - L_1 \times sin\alpha)\tag{1}$$

where $\alpha$ is the angle between the centerline of the lower pull rod and the horizontal plane.

　　　As the tractor's three-point hitch mechanism and the subsoiler are rigidly connected, the angle $\beta$ between the tractor's forward direction and the horizontal plane is regarded as the slope angle of the operating plot when the tractor is operating in the tillage operation with a slope greater than 2°. The attitude rotation angle is specified, and the clockwise direction is positive, that is, the slope angle is positive for upslope and negative

for downslope, as shown in Figure 3a,b. The following formula can be derived from the geometric relationship:

$$\begin{cases} \theta = \alpha - \beta \\ H = H_2 - (H_1 - L_1 \times sin\theta) \end{cases} \quad (2)$$

where $\theta$ is the angle between the center line of the lower pull rod and the working ground of the tractor unit, $\alpha$ is angle between the center line of the lower pull rod and the horizontal plane, and $\beta$ is the angle between the tractor's forward direction and the horizontal plane.

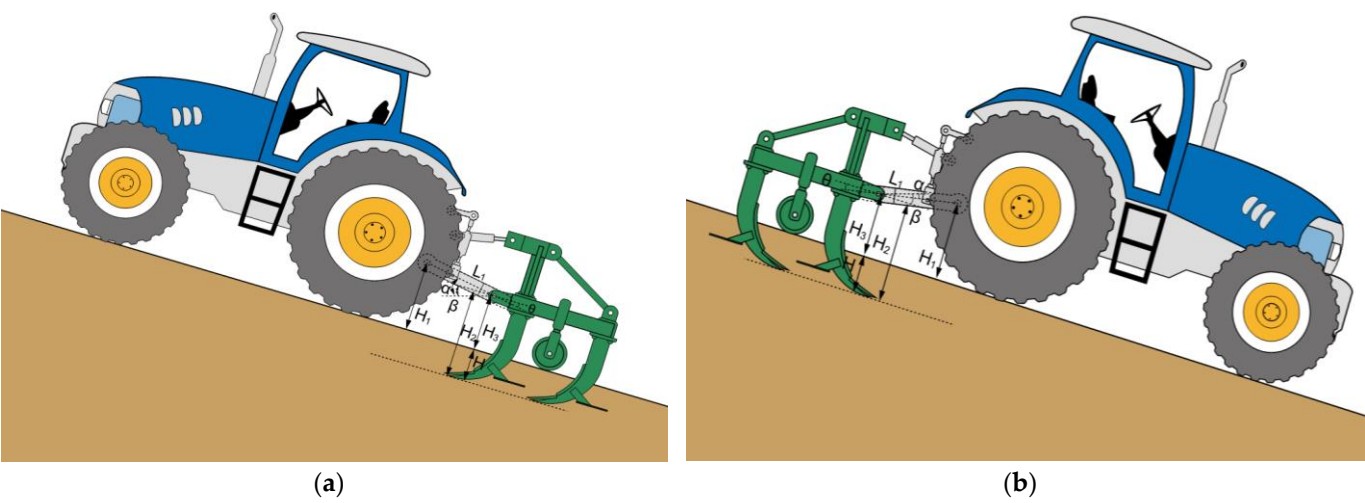

(**a**)  (**b**)

**Figure 3.** Tillage depth detection of suspended subsoiler: (**a**) uphill operation and (**b**) downhill operation.

### 3.2. Detection Method of Corn Straw Coverage Rate

Considering that straws in agricultural fields have various forms, and delicate straws are difficult to identify accurately, this study proposed a corn straw coverage rate detection algorithm based on K-means clustering according to machine vision technology. This method effectively solves the problem of traditional manual measurement methods, which are greatly influenced by subjective factors and have, thus, a low efficiency. Figure 4 shows the steps of the detection algorithm, which includes reading the original image of corn straw, denoising the image with a Gaussian filter, graying the image to separate straw from the soil background, calculating the number of soil background and straw pixel points based on the K-means clustering method, and finally, obtaining the calculation result of the corn straw coverage rate.

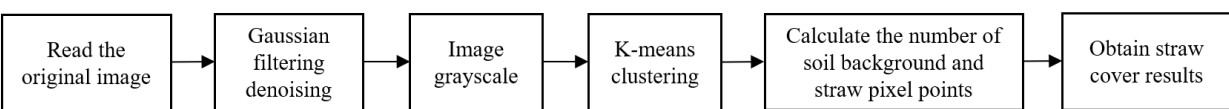

**Figure 4.** Flow chart of corn straw coverage rate detection.

K-means clustering is a classical algorithm in machine learning unsupervised classification. There are two problems in the K-means clustering process: (1) the number of clusters and (2) the cluster center [19,28,29]. In this study, a camera was installed at the front of the tractor, and the corn straw images were taken by tilting the camera towards the ground. In the obtained corn straw images, the target and background were distinguishable. The target was straw, and the background was soil, so the number of clusters could be determined as 2. The initial clustering center was determined by a random method. Then, the distance between each sample point and the clustering center was calculated separately, and iteration was carried out after the new clustering center was determined. The clustering iteration continued until the clustering center no longer changed, meaning

that the classification of straw and soil was completed. The corn straw coverage rate was calculated using Formula (3).

$$PC = \frac{C_{straw}}{C_{straw} + C_{soil}} \times 100\% \tag{3}$$

where *PC* is the detection value of corn straw coverage rate, $C_{straw}$ is the number of straw pixels, and $C_{soil}$ is the number of soil pixels.

To evaluate the detection algorithm of the corn straw coverage rate, the correlation coefficient and relative error were adopted. The correlation coefficient was used to evaluate the correlation between the results of corn straw coverage rate detection and the results of manual image labeling; the relative error was used to judge the error of this detection algorithm. The correlation coefficient and relative error were obtained by Formulas (4) and (5), respectively.

$$r = \frac{\sum_{k=1}^{n}(x_{1k} - \overline{x_1})(x_{2k} - \overline{x_2})}{\sqrt{\sum_{k=1}^{n}(x_{1k} - \overline{x_1})^2 \sum_{k=1}^{n}(x_{2k} - \overline{x_2})^2}} \tag{4}$$

where *r* is correlation coefficient, $x_1$ is standard values of corn straw coverage rate, and $x_2$ is detection value of corn straw coverage rate.

$$E = \frac{PC - F}{F} \times 100\% \tag{5}$$

where *E* is the relative error, *PC* is the detection value of corn straw coverage rate, and *F* is the standard value of corn straw coverage rate.

The camera installed on the front of the tractor was set at 45° for image capture. Before each operation, the lens was wiped once. Shading components were installed on the front of the camera to ensure that the collected images were not affected by dust, bright light, and other factors.

In the experimental process, to evaluate the accuracy of the results, the truth-value acquisition method of straw cover rate used in this paper was to mark and identify the images manually. Various shaped and tiny straws were accurately marked manually. The image was segmented by the binarization method, and the pixel points were calculated by a computer. This result was used as the truth-value of straw cover rate in the field to evaluate the accuracy of this algorithm. This truth-value acquisition method relies on manual labeling of the image, and the results were more accurate. Moreover, the images processed by the truth-value acquisition method could be guaranteed to correspond precisely with the images processed by the algorithm in this paper. The accurate matching of samples was achieved. Finally, the test results were evaluated by the correlation coefficient and mean error.

## 4. Research and Development of a Remote Monitoring System for Agricultural Machinery Operation in Conservation Tillage

### 4.1. Operation Monitoring Terminal

The operation monitoring terminal is the main component of the tillage depth online detection device and image data acquisition and processing. The terminal consists of a control and processing module, global navigation satellite system (GNSS) positioning module, 4G communication module, display and alarm module, storage module of data, etc. The control and processing module uses an ARM Cortex-M4 microprocessor (STMicroelectronics, Geneva, Switzerland), which receives and processes the data collected by the tillage depth detection device and the corn straw coverage rate detection. Based on this module, information, such as GNSS, tractor working conditions, and agricultural equipment structure, is integrated. The operation information is uploaded to the monitoring platform in real-time through the 4G network to complete the remote online monitoring of the operation. The terminal has a built-in trans-flash (TF) card and flash storage, which can save the operation and image data in real time. Moreover, the terminal is integrated with a

display and alarm module, which will warn the driver when the operation quality is not qualified or the equipment is malfunctioning. The main hardware structure is shown in Figure 5.

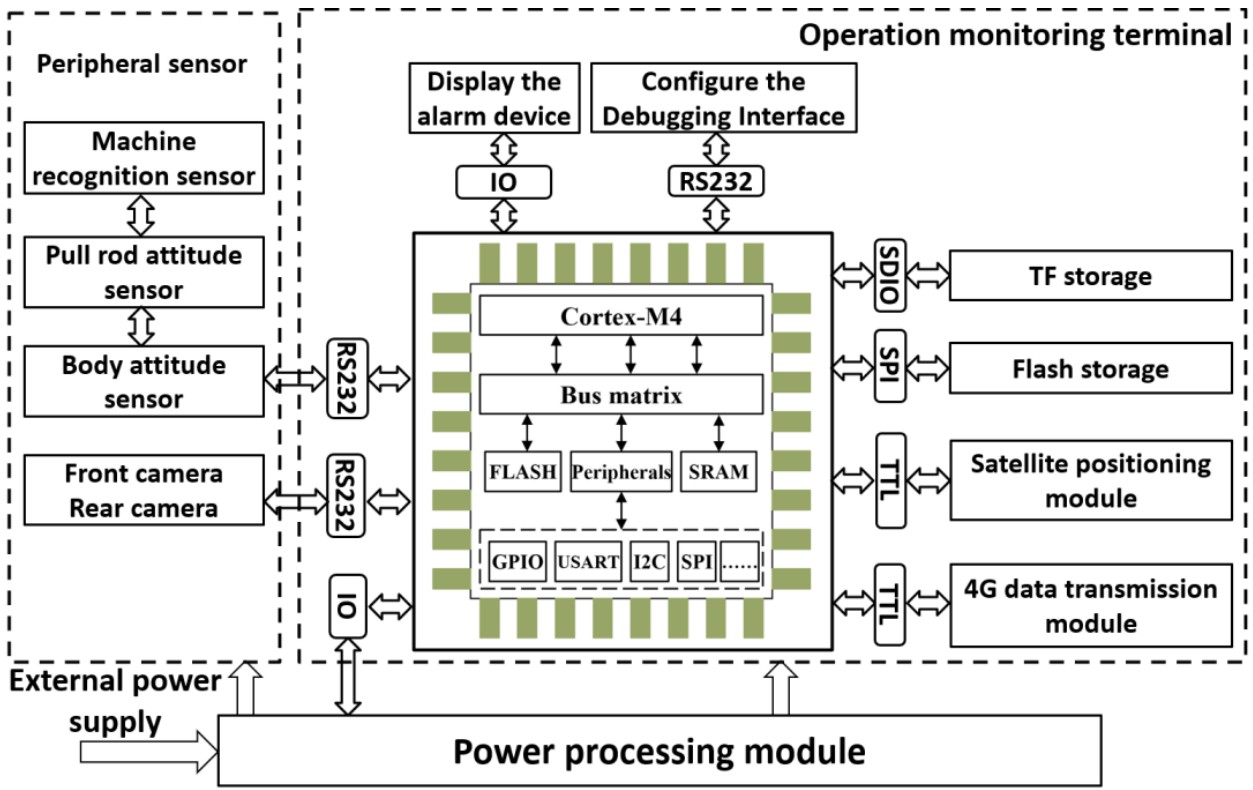

**Figure 5.** Hardware structure of operation monitoring terminal.

The monitoring terminal is mainly responsible for the real-time online monitoring of the sensing, processing, and transmission of operational information. The sampling frequency is 50 Hz. It can realize functions such as timely acquisition, storage, and operation information analysis. The terminal's functions mainly includes the driver of each hardware module, the reception and processing of GNSS data, processing and analysis of tillage depth data, acquisition and processing of straw image data, transmission of 4G wireless network data, and storage of local data. Among them, the data transmission of mobile communication is greatly affected by the signal; especially in the environment of field operation, there will often be no or weak signal, so the integrity of data transmission is the focus of the whole system. When the network signal is weak or interrupted, the current data are stored to avoid data loss during transmission. When the network signal returns, the saved data will be uploaded to the service system for supervision and management of remote operations, usually using the FIFO method. The main software flow of the monitoring terminal is shown in Figure 6.

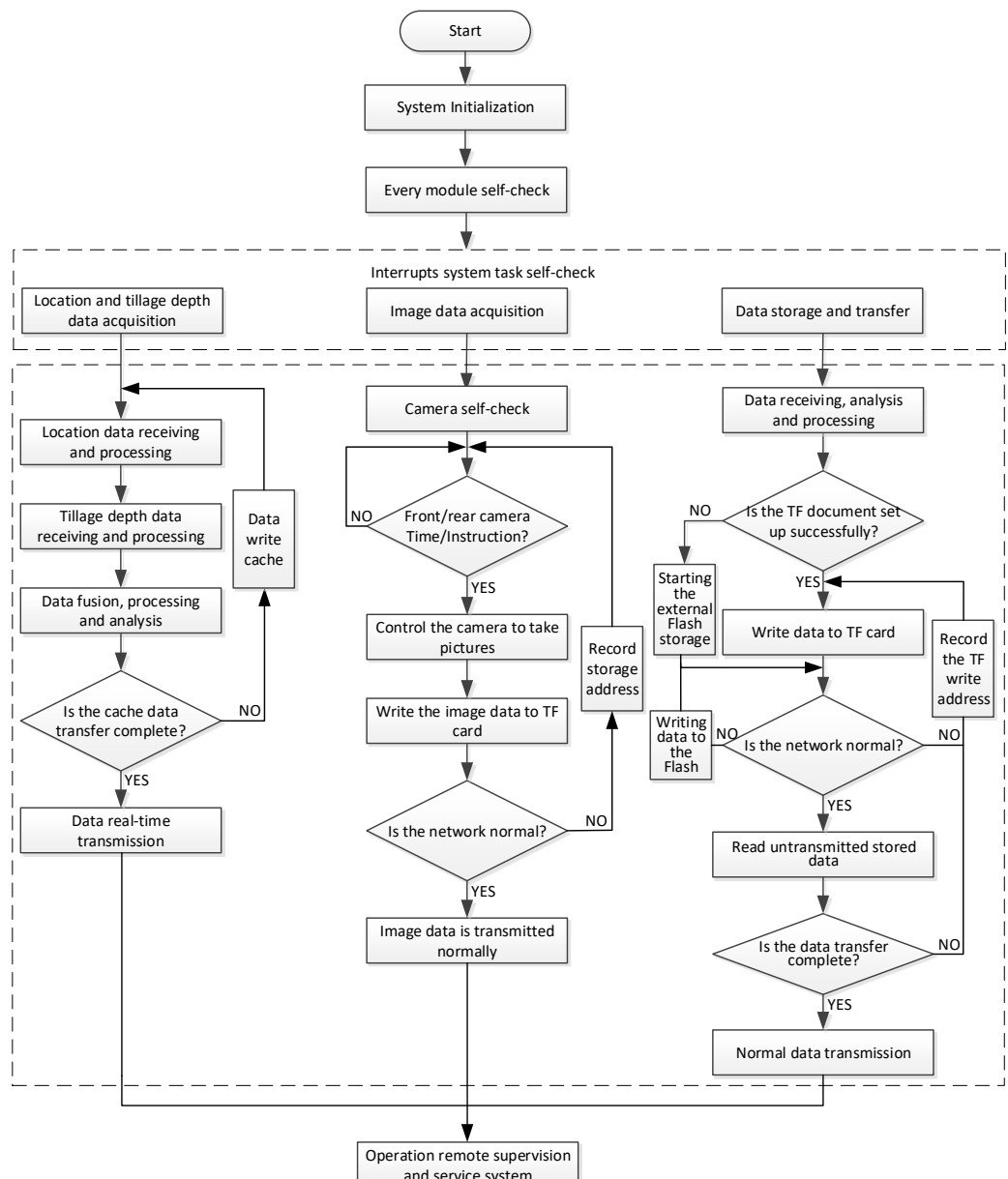

**Figure 6.** Schematic diagram of the implementation process of the terminal function.

### 4.2. A Service System for Remote Monitoring

The system was built using service-oriented architecture (SOA) technology and divided into infrastructure, data, service, business logic, and application presentation layers. The system provides access to business data for managing agricultural machinery, farmers, and agricultural machinery cooperatives in a browser/server mode. The overall architecture of the system is shown in Figure 7.

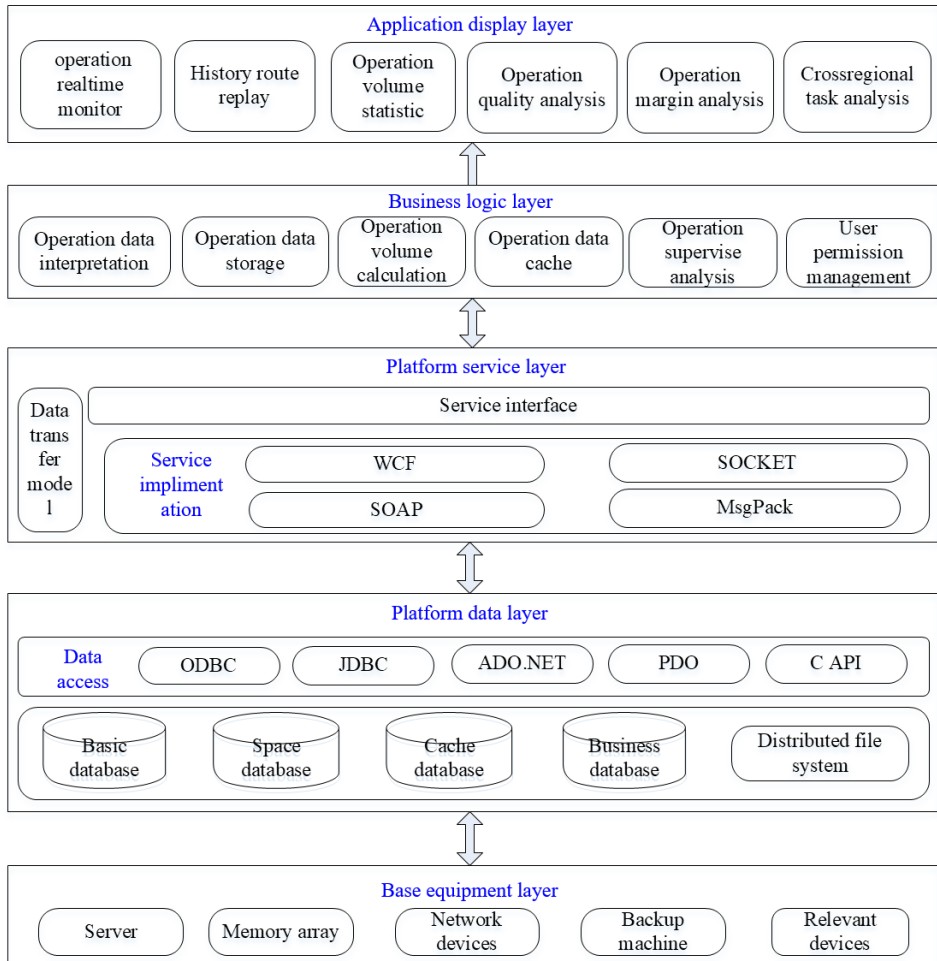

**Figure 7.** Overall architecture of the system.

The infrastructure layer consists of servers, storage disk arrays, network devices, backup all-in-one machines, and system basic software. On the storage side of the basic data, the technology of disk array storage architecture is used to realize the storage of the basic data of users. On the backup side of the platform data, the technology of backup integration architecture is adopted. By storing and backing up critical data, such as basic databases, files, and operating systems, it meets the needs of data backup and disaster recovery.

The data layer consists of a base database cluster, a geographic information spatial database cluster, and a cache database cluster. The base database cluster adopts the cluster building scheme of MySQL Replication technology. It can meet the service requirements of agricultural machinery operation data storage and business data access. The PostGIS-based spatial database cluster can realize data storage functions, such as of agricultural machinery operation plots, operation locations, operation areas, operation omission areas, and operation overlapping areas. The cache database cluster was constructed in the Redis Cluster mode to meet data access needs during peak agricultural machinery operations.

The service layer contains two segments. On the one hand, it realizes the function of sending and receiving operation data and data queue buffering based on Socket technology. On the other hand, based on windows communication foundation (WCF) technology, the business interface and data transfer object (DTO) model are designed for modules such as the management of basic data, statistics of agricultural machinery operations, real-time monitoring of remote operations, playback of farm machinery operation trajectories, analysis of operation quality, analysis of operation area boundaries, and analysis of cross-area operations and management of user rights.

At the business logic level, the business interface based on the service layer realizes business functions such as parsing of agricultural machinery operation data, calculation of operation volume, entry of operation data, analysis of operation supervision, caching of operation data, encryption of data, compression of data, and map services.

The presentation layer mainly includes functional modules, such as real-time monitoring of operations, playback of historical trajectories, statistics of operation quantity, analysis of operation quality, analysis of operation area boundaries, and analysis of cross-area operations. The platform is rapidly deployed, upgraded, and maintained in B/S mode.

## 5. Experiments and Results Analysis

### 5.1. Tillage Depth Measurement Test and Result Analysis

To test the reliability and stability of the tillage depth detection system of the suspended subsoiler, a static calibration experiment was carried out in the Xiaotangshan National Experiment Station for Precision Agriculture, Changping District, Beijing, in August 2021. A Dongfanghong 1104 tractor (China YTO Group Co. Ltd., Luoyang, China) and Dahua Baolai 1S-230 subsoiler (Dahua Machinery Co., Ltd., Jining, China) were selected as test subjects. In October 2021, a Case 210 tractor (CNHI, New York, NY, USA) and 1S-360 subsoiler were used in 597 Farm, Baoqing County, Shuangyashan City, Heilongjiang Province, for field stability experiments. The experimental site is shown in Figure 8.

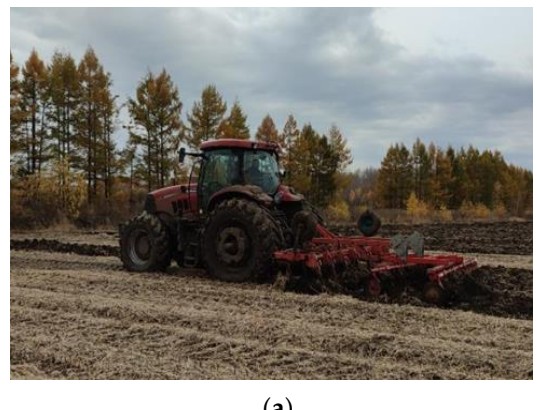 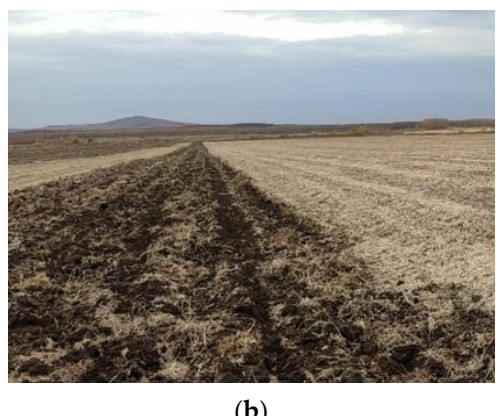

(**a**)                                   (**b**)

**Figure 8.** Field experiments at 597 Farm: (**a**) testing site and (**b**) the condition of the soil after tillage.

(1)    Error analysis of tillage depth detection at static state

The subsoiler unit was calibrated on the horizontal ground according to the detection model. Then, the tractor was parked on the slope at about 6°, and the tillage depth of the subsoiler was adjusted from small to large and then from large to small at 5-cm intervals in 10–35 cm sections. The output results of the detection system when going uphill and downhill were compared with the manual measurement data. The results of static tillage depth detection during uphill and downhill are shown in Tables 1 and 2, Figures 9 and 10.

**Table 1.** Static test results of tillage depth detection of the suspended subsoiler when going uphill.

| Sample | Slope Angle (°) | System Measurement (cm) | Manual Measurement (cm) | Relative Error (%) |
|---|---|---|---|---|
| 1 | 5.21 | 12.30 | 10.90 | 1.40 |
| 2 | 6.02 | 15.48 | 15.50 | −0.02 |
| 3 | 5.05 | 20.78 | 20.30 | 0.48 |
| 4 | 5.28 | 24.69 | 25.30 | −0.61 |
| 5 | 5.76 | 30.37 | 30.60 | −0.23 |
| 6 | 6.53 | 34.90 | 35.40 | −0.50 |
| 7 | 5.79 | 35.14 | 35.80 | −0.66 |

**Table 1.** *Cont.*

| Sample | Slope Angle (°) | System Measurement (cm) | Manual Measurement (cm) | Relative Error (%) |
|---|---|---|---|---|
| 8 | 5.88 | 30.17 | 30.70 | −0.53 |
| 9 | 6.14 | 25.70 | 25.20 | 0.50 |
| 10 | 5.14 | 19.51 | 20.40 | −0.89 |
| 11 | 5.39 | 15.62 | 15.40 | 0.22 |
| 12 | 5.41 | 9.47 | 10.50 | −1.03 |

**Table 2.** Static test results of tillage depth detection of the suspended subsoiler when going downhill.

| Sample | Slope Angle (°) | System Measurement (cm) | Manual Measurement (cm) | Relative Error (%) |
|---|---|---|---|---|
| 1 | −6.95 | 11.60 | 10.30 | 1.30 |
| 2 | −6.94 | 16.02 | 15.70 | 0.32 |
| 3 | −7.26 | 20.58 | 20.40 | 0.18 |
| 4 | −7.06 | 25.33 | 25.30 | 0.03 |
| 5 | −7.09 | 31.05 | 30.40 | 0.65 |
| 6 | −6.66 | 35.67 | 35.80 | −0.13 |
| 7 | −6.78 | 35.53 | 35.10 | 0.43 |
| 8 | −6.41 | 30.66 | 30.20 | 0.46 |
| 9 | −6.59 | 26.42 | 25.40 | 1.02 |
| 10 | −7.09 | 19.91 | 20.90 | −0.99 |
| 11 | −6.70 | 16.10 | 15.20 | 0.90 |
| 12 | −6.92 | 10.21 | 10.10 | 0.11 |

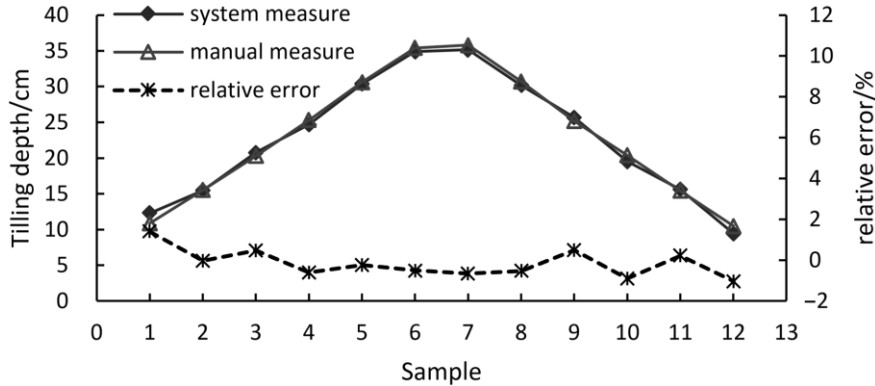

**Figure 9.** Results analysis of tillage depth detection test of the suspended subsoiler when going uphill.

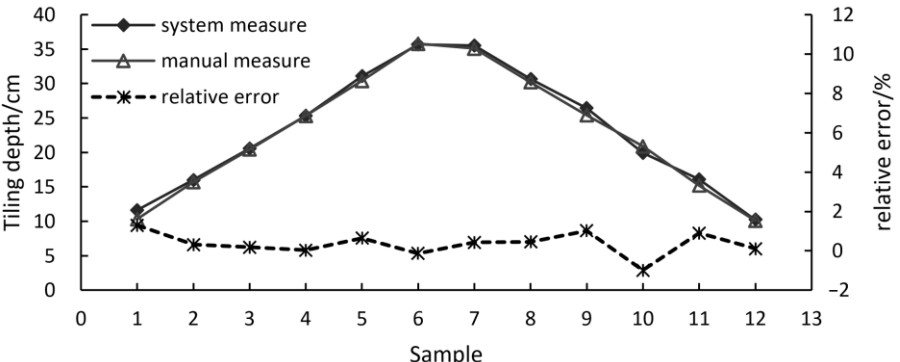

**Figure 10.** Results analysis of tillage depth detection test of the suspended subsoiler when going downhill.

The static average, maximum, minimum, and root means square error of the tillage depth detection system for the suspended subsoiler when going up the slope were 0.16, 1.40, 0.02, and 0.67 cm, respectively. The downhill static average error was 0.36 cm, the maximum error was 1.30 cm, the minimum error was 0.03 cm, and the root means square error was 0.57 cm.

(2)　Results analysis of tillage depth stability

Through the hydraulic suspension mechanism, the tractor dropped the subsoiler to the same fixed position during operation. On the 2° and 4° slopes, the tractor operated a stroke in the forward and return directions to measure the tillage depth data in real-time. A field experiment with 28-cm tillage depth was used as an example to demonstrate the process of signal processing. The sampling rate was 50 Hz. The measured data were converted and processed to obtain the measured data of tillage depth, as shown in Figure 11.

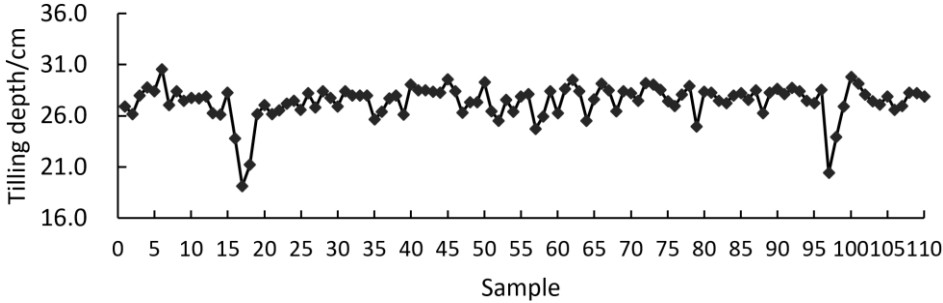

**Figure 11.** Measured data of tillage depth.

To ensure the usability and accuracy of the data, removal of the singular value, compensation of the singular data difference, and sliding average filtering were performed, as shown in Figure 12.

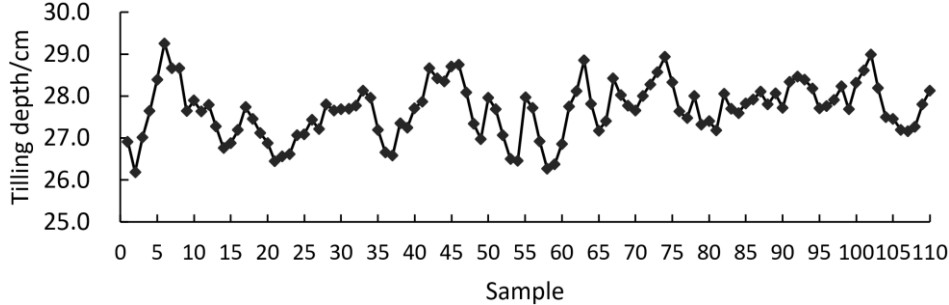

**Figure 12.** Tillage depth data after processing.

According to the actual tillage requirements, the tillage depths for this trial were set at about 28 and 33 cm, respectively. The results of the experiment with the tillage depth set at about 28 cm are shown in Figure 13.

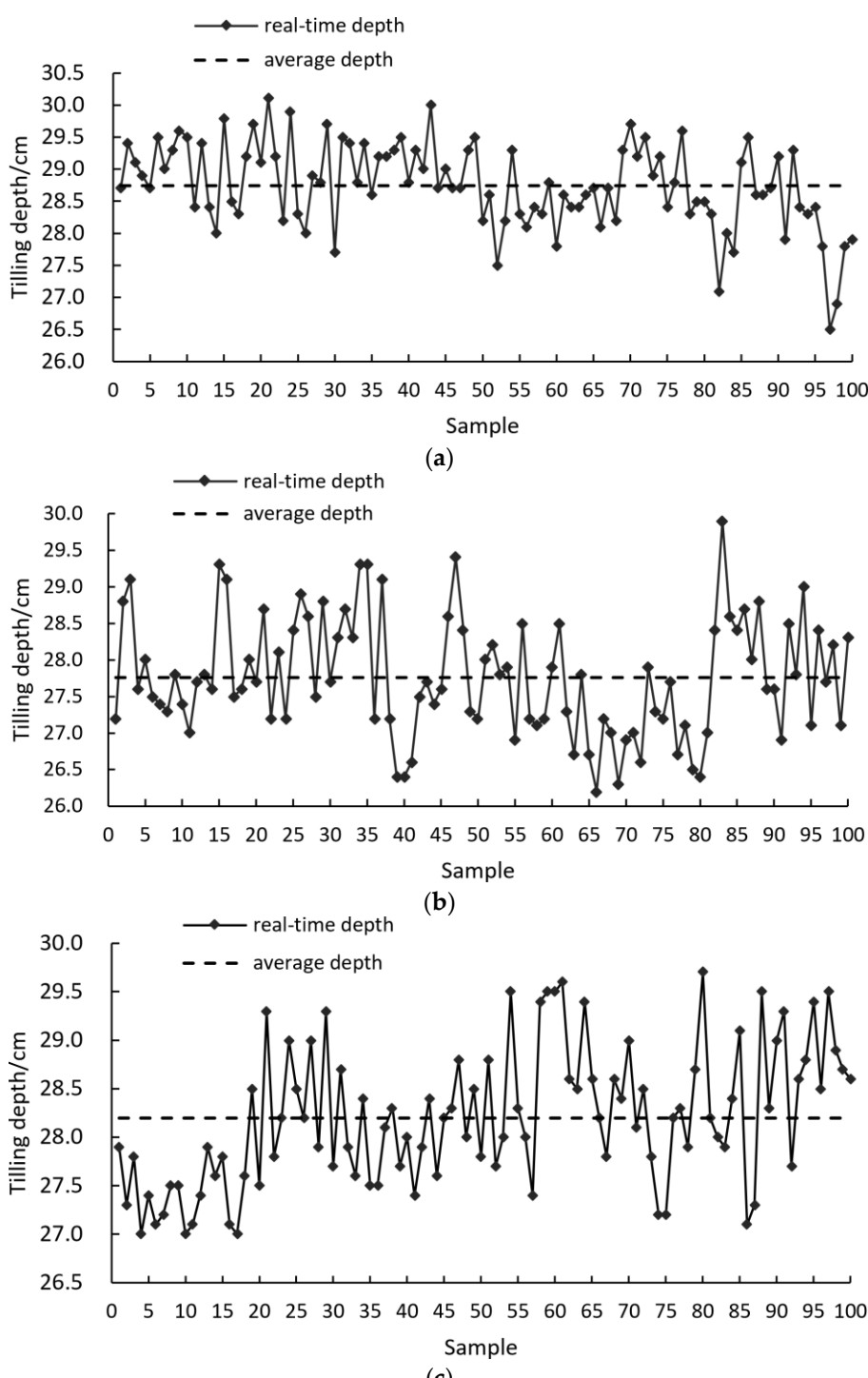

**Figure 13.** *Cont.*

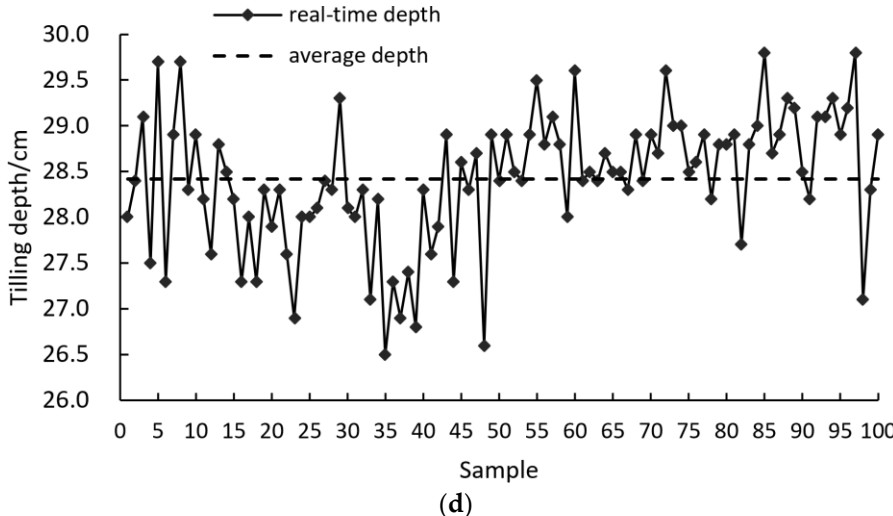

(**d**)

**Figure 13.** Field test results of the tillage depth detection system under different slopes: (**a**) uphill slope angle is 2°, working speed is 2.80 m/s; (**b**) downhill slope angle −2°, working speed 2.97 m/s; (**c**) uphill slope angle 4°, working speed 2.76 m/s; (**d**) downhill slope angle −4°, working speed 2.78 m/s.

The analysis of Figure 13 shows that the average tillage depth measured by the total stroke system was 28.28 cm, the average tillage depth measured manually was 27.73 cm, the mean error during dynamic tillage depth measurement was 0.55 cm, the root mean square error was 0.64 cm, the variation coefficient of tillage depth was 2.87%, and the stability coefficients of tillage depth for each stroke were 97.64%, 97.08%, 97.46%, and 97.44%, respectively.

The results of the experiment with the tillage depth set at about 33 cm are shown in the Figure 14.

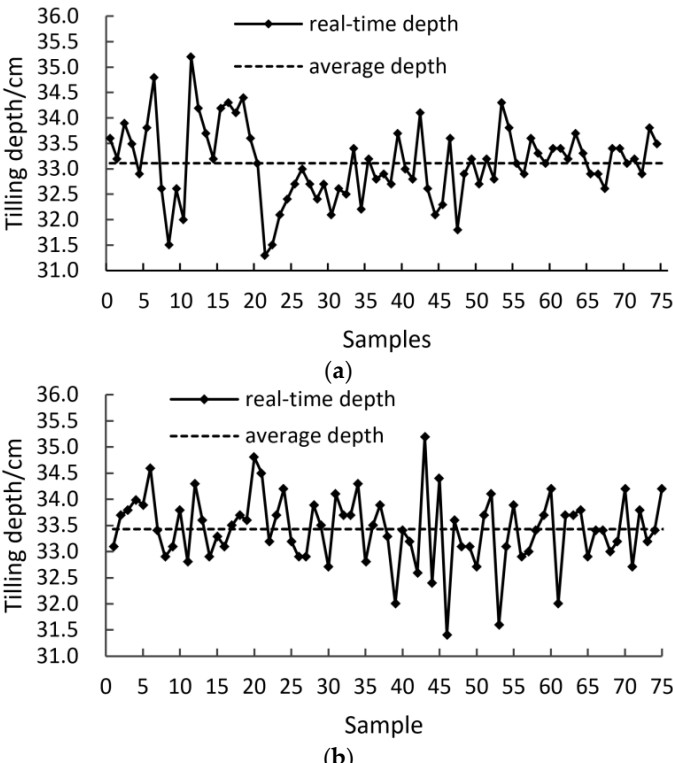

(**a**)

(**b**)

**Figure 14.** *Cont.*

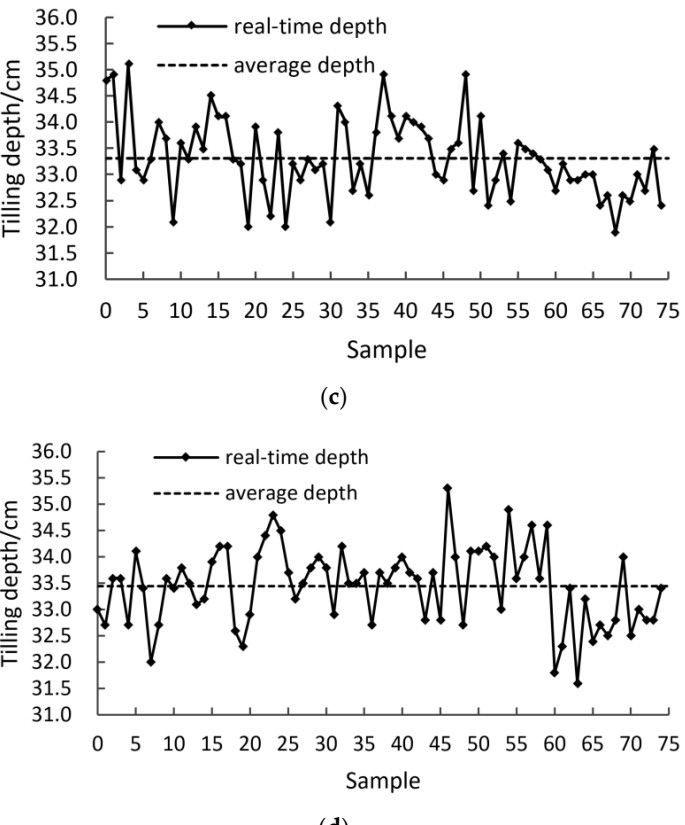

**Figure 14.** Field test results of tillage depth detection system under different slopes: (**a**) uphill slope angle 2°, working speed 2.19 m/s; (**b**) downhill slope angle −2°, working speed 2.44 m/s; (**c**) uphill slope angle 4°, working speed 2.62 m/s; (**d**) downhill slope angle −4°, working speed 2.37 m/s.

Figure 14 shows that the average tillage depth measured by the total stroke system was 33.32 cm; the average tillage depth measured manually was 32.71 cm; the mean error during dynamic tillage depth measurement was 0.61 cm; the root mean square error was 0.73 cm; the variation coefficient of tillage depth was 2.21%; and the stability coefficients of tillage depth for each stroke were 97.75%, 97.95%, 97.77%, and 97.79%, respectively.

Therefore, the results of the comprehensive field experiments show that under complex terrains, such as unfluctuating farmland slope, weeds, crop residues, and ridges, the average error and root mean square error of static tillage depth detection were 0.36 and 0.62 cm, respectively. The average values of dynamic measurement errors were 0.55 and 0.61 cm, and the root mean square errors were 0.64 and 0.73 cm for the 28 and 33 cm tillage depth tests, and the variation coefficients of tillage depth did not exceed 3%. The calculation results of the proposed method were basically the same as the average error of 0.45 of the method proposed by Yin [9]; however, the validation test of the proposed method considered a larger field slope, and the device was simpler and easier to apply. Therefore, the tillage depth measurement method based on the body and the lower pull rod of the three-point hitch mechanism and the technology of dual attitude integration and sensor data processing could effectively solve the problem of slope tillage depth detection.

*5.2. Corn Straw Cover Rate Detection Test and Result Analysis*

In May 2022, 45 images were collected for the actual field validation of this algorithm in corn plots in Northeast China. The manual image tagging method was used to obtain the standard values of straw cover rate, which ranged from 5% to 85%, with the low straw cover rate (5–30%), medium straw cover rate (30–60%), and high straw cover rate (60–85%). In addition, the validation images included various sceneries. The identification of different gradients of the straw cover rate is shown in Figure 15.

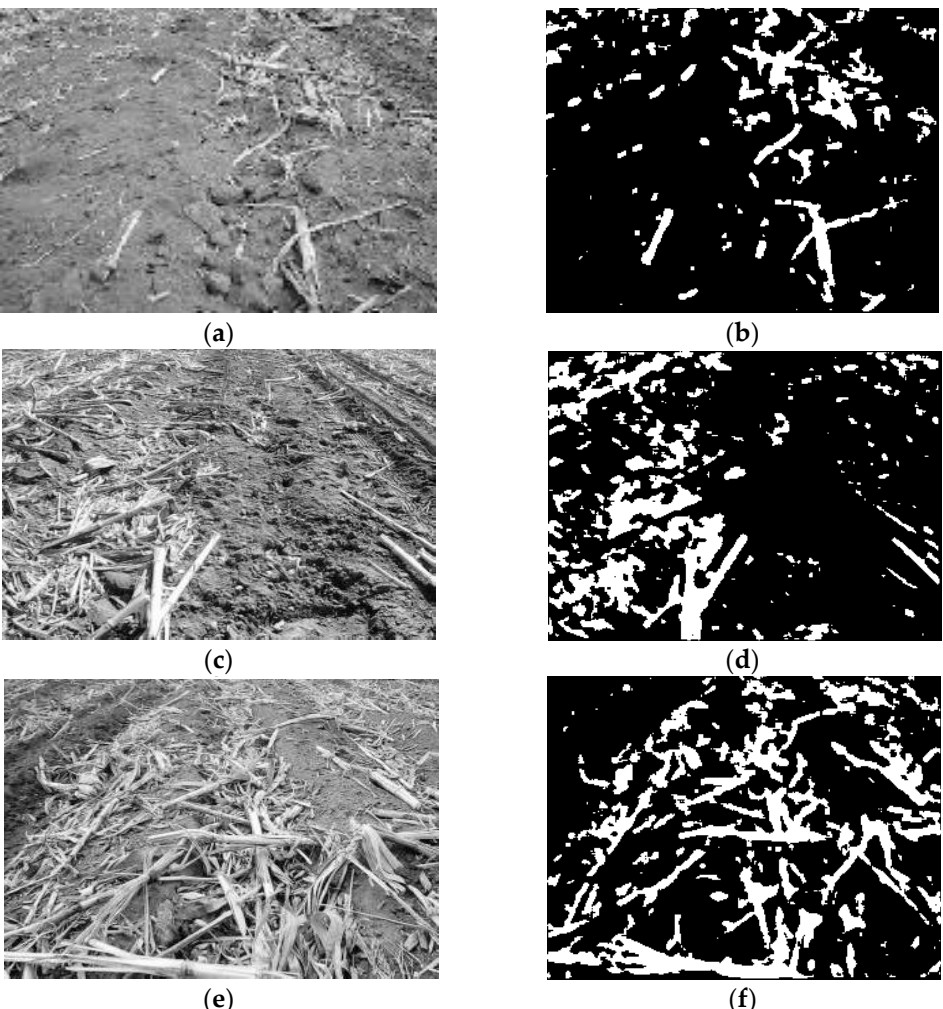

**Figure 15.** Recognition effect of straw coverage rate with different gradients: original image of (**a**) low straw coverage rate; (**c**) medium straw coverage rate; (**e**) high straw coverage rate and algorithm recognition results of (**b**) low straw coverage rate; (**d**) medium straw coverage rate; (**f**) high straw coverage rate.

The straw cover rate detection results corresponding to the experiment's three levels were 9.66%, 31.35%, and 71.90%, which could accurately identify the straw and soil background. Correlation coefficient diagram of the straw coverage rate algorithm and artificial image labeling method is shown in Figure 16.

In summary, the correlation coefficient of the proposed algorithm and manual image tagging method for testing and validating different gradients of straw cover rate from 5% to 85% reached 0.92. The average error predicted by the proposed method was 9.69%, which was comparable to the 90% accuracy of the method proposed by Li et al., meeting the requirements for the rapid detection of the straw cover rate in the field [17,18]. In addition, the sources of error generation mainly included maize stalks, leaves, and rhizomes with different morphologies and location distributions. Particularly, maize rhizomes, which easily produce shadows, which are easily misidentified as straw classes, cause errors in straw and soil segmentation.

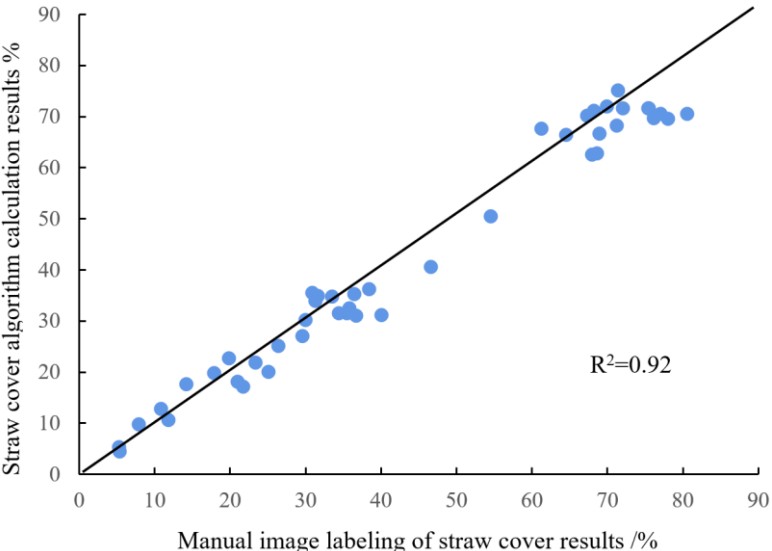

**Figure 16.** Correlation coefficient diagram of the straw coverage rate algorithm and artificial image labeling method.

### 5.3. Main Functions Verification of the Remote Monitoring System

This section demonstrates the main functions of the remote monitoring system for conservation tillage agricultural machinery operation, which mainly includes the parts of real-time operation monitoring, track playback, operation quantity analysis, and operation quality analysis, as shown in Figure 17.

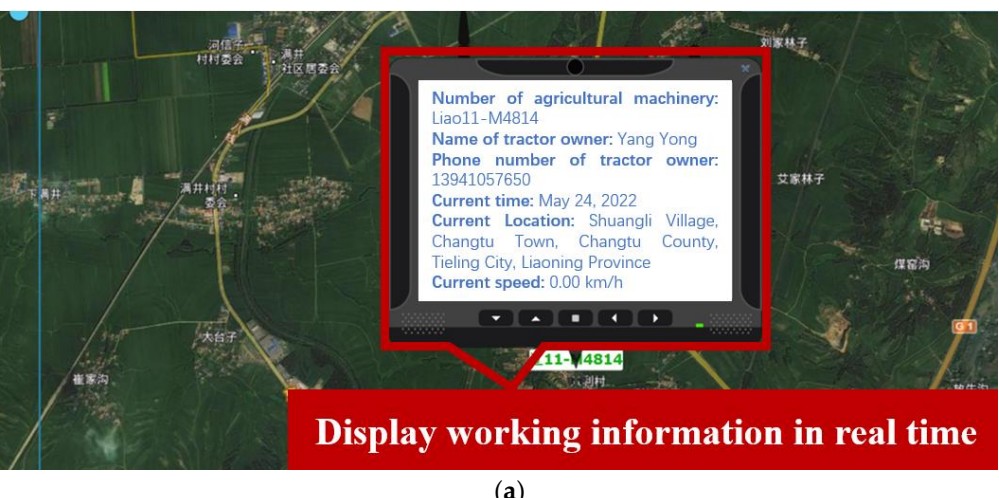

(**a**)

**Figure 17.** *Cont.*

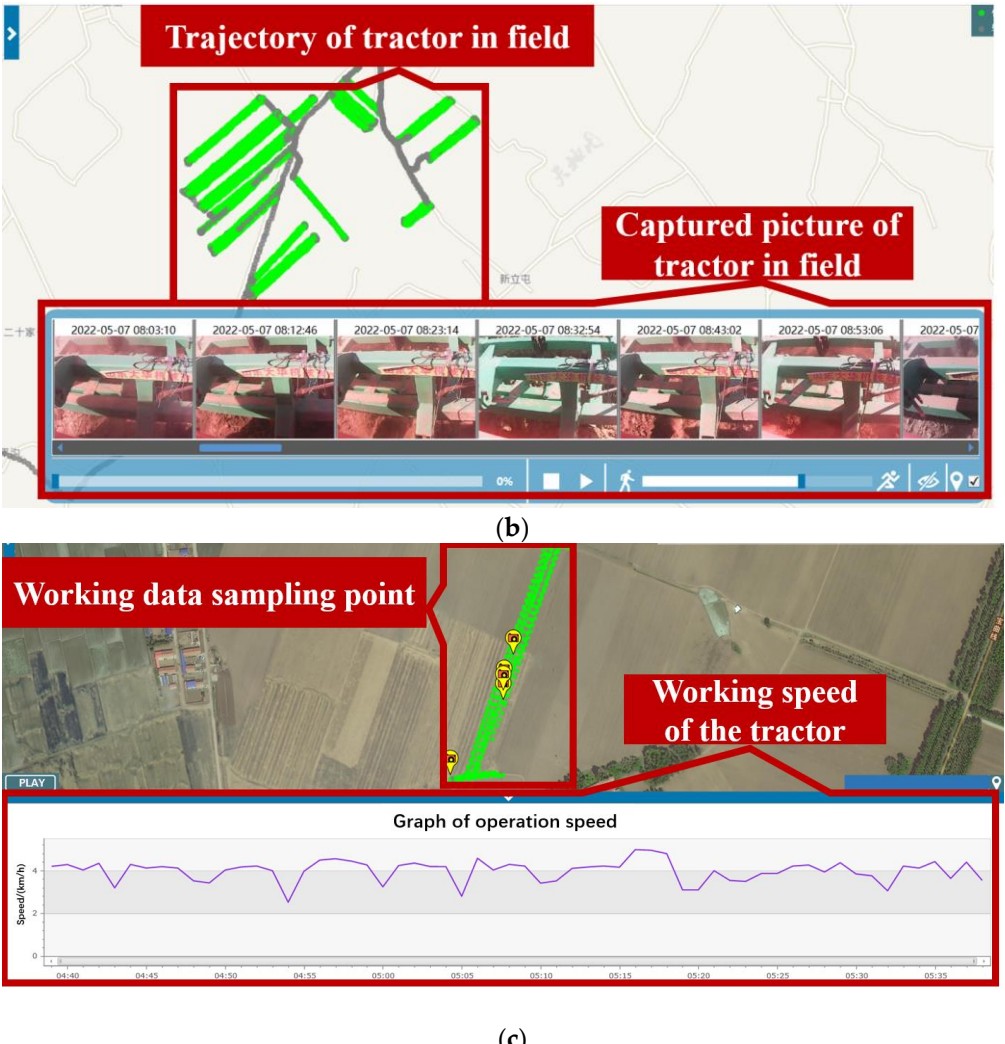

**Figure 17.** Main interface of the remote monitoring system for conservation tillage agricultural machinery operation: (**a**) real-time monitoring interface; (**b**) trajectory playback interface; (**c**) working analysis interface.

Figure 17a shows the machine operation position, operation task, and operation status of conservation tillage with an information update frequency of 0.5 Hz, which meets the requirement of real-time monitoring of conservation tillage. Figure 17b shows the machine operation trajectory for typical working conditions of conservation tillage and the captured images to assist in analyzing the operation status. Figure 17c shows the machine operation speed of conservation tillage and the image map's distribution of sampling points.

For the operation quality analysis module of the remote monitoring system of conservation tillage, the tillage depth detection module was illustrated by a John Deere 554 tractor (John Deere Investments Ltd., Moline, IL, USA)with the license plate "Liao 11-40528", and the straw cover rate detection module was illustrated by a Lovol 904 tractor (Weizhai Lovol Heavy Industry Co., Ltd., Weifang, China) with the license plate "Liao 11-50024", as shown in Figure 18.

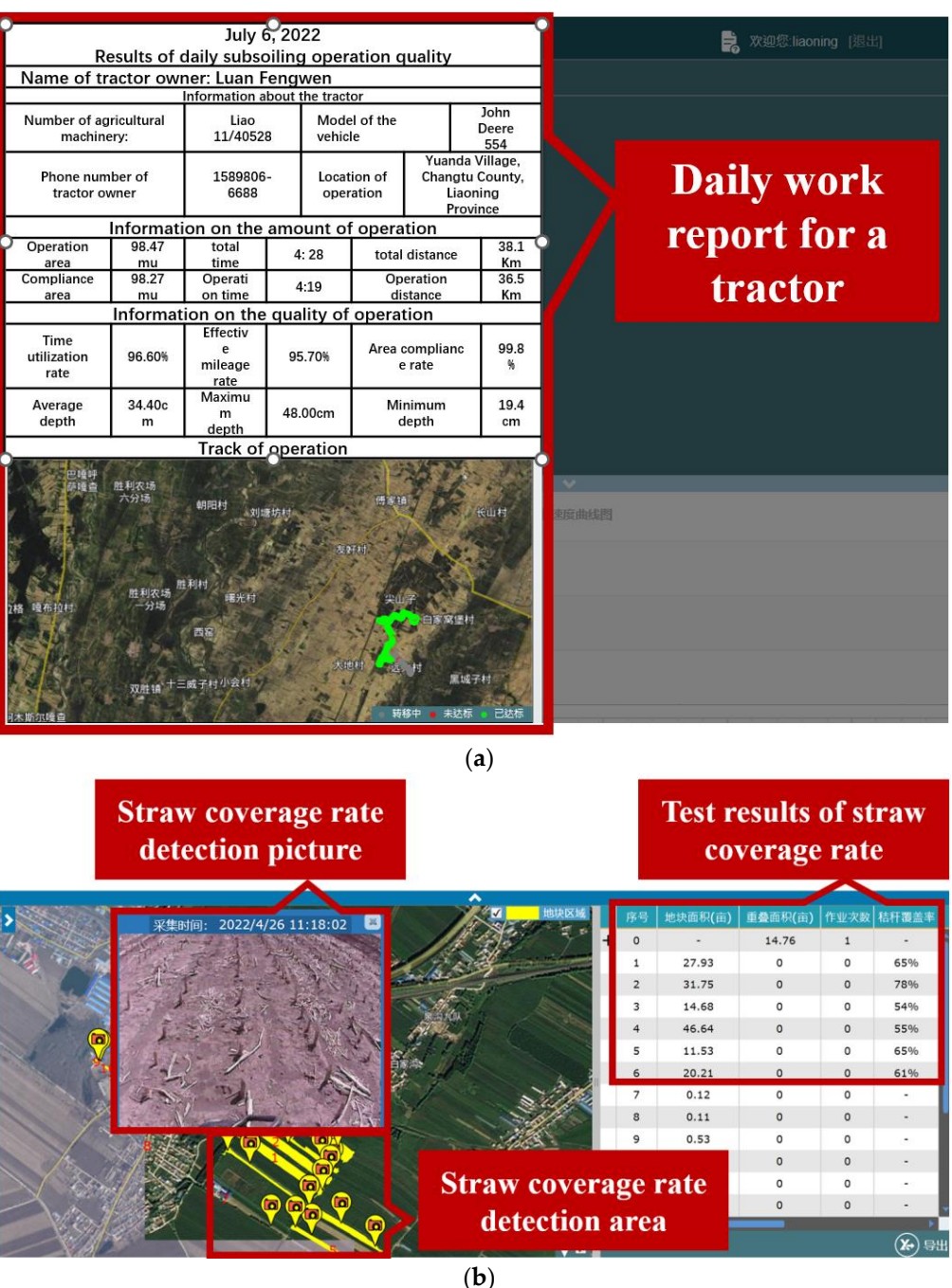

**Figure 18.** Operation quality analysis module of a remote monitoring system for conservation tillage agricultural machinery operation: (**a**) analysis results of tillage depth detection and (**b**) analysis results of corn straw coverage rate.

The operation quality analysis module of a remote monitoring system of conservation tillage agricultural machinery mainly included vehicle information, operation quantity information, operation quality information, and trajectory information. The green dots in the trajectory information represent tillage quality that meets the agronomic requirements, the red trajectory dots represent tillage quality that does not meet the agronomic requirements, and the gray trajectory dots represent no operation.

Figure 18a shows the data for the total subsoiling operation area, effective subsoiling operation area, and average tillage depth on 6 July 2022. The machine had a total subsoiling operation area of 98.47 ha, an effective subsoiling operation area of 98.27 ha, an area

compliance rate of 99.8%, an average tillage depth of 34.40 cm, and an operation time of 4 h and 19 min on that working day. Figure 18b shows the total straw return area, effective straw return area, and straw cover rate on 26 April 2022. The machine had an area of 253.51 ha and an effective operating area of 238.75 ha, and the straw coverage of the plots ranged from 55–65%. Compared with the manual test, the accuracy of the analysis method was higher. Therefore, the remote monitoring system of agricultural machine operation constructed in this paper is oriented to conservation tillage. It realizes the information monitoring of operation quality and quantity and solves the problems of low efficiency and incomprehensiveness of traditional manual monitoring.

## 6. Conclusions

Aiming at the remote monitoring demand of agricultural machinery operation quality for conservation tillage, this paper developed a remote monitoring system of agricultural machinery operation for conservation tillage. The system included a tillage depth detection device, a straw cover rate detection unit, an operation monitoring terminal, and a remote operation supervision service system. The tillage depth detection device and straw cover rate detection unit adopted multi-attitude sensor fusion and machine vision methods to realize tillage depth and straw cover rate detection, respectively. The operation monitoring terminal adopted a high-performance embedded processor to realize multi-source information collection, analysis, storage, and remote transmission. The remote operation monitoring service system adopted service-oriented technology architecture. The system was oriented to conservation tillage and realized the information monitoring of operation quality and quantity, solving the problems of low efficiency and incomprehensiveness of traditional manual monitoring.

This paper systematically studied the operation quality analysis method of subsoiling operation and straw return in conservation tillage. Aiming at the problem that tillage depth detection in a slope environment is easily affected by attitude change and vibration, and the accuracy is difficult to guarantee, this paper proposed a tillage depth measurement method based on a body and three-point hitch mechanism lower pull rod of the tractor. This method integrated dual attitude fusion and sensor data processing to solve the subsoiling operation tillage depth detection in slope. For corn straw cover rate detection, an image processing algorithm based on K-means clustering was proposed, which solved the problem of subjective factors with significant influence and low efficiency of traditional manual measurement methods. Field tests showed that the system had a static mean error and root mean square error of 0.16 and 0.67 cm for uphill and 0.36 and 0.57 cm for downhill; the mean dynamic measurement errors of the 28 and 33 cm tillage depth tests were 0.55 and 0.61 cm, with root mean square errors of 0.64 and 0.73 cm, respectively. The coefficients of variation of tillage depth did not exceed 3%. The calculation results of the proposed method were basically the same as the average error of 0.45 of the method proposed by Yin [9]. The correlation coefficient between the straw cover rate detection algorithm and this paper's manual image marking method reached 0.92, with an average error of 9.69%, which is comparable to the 90% accuracy of the method proposed by Li et al. in the existing study [17,18]. The remote monitoring service system of agricultural machinery operation has the functions of real-time operation monitoring, historical track playback, operation area statistics, operation quality analysis, and operation area boundary analysis. The detection accuracy can meet the requirements of real-time online monitoring of tillage depth and straw coverage rate under complex farming operation environments and has good anti-interference ability.

The conservation tillage agricultural machinery operation monitoring terminal and service system developed in this paper has been widely applied in Northeast China, realizing the remote monitoring of subsoiling and straw returning and forming the application mode of "Internet + agricultural machinery operation." This research has played an essential role in ensuring the quality of conservation tillage operations, improving the standardization level of agricultural machinery operations and the protection and sustainable use

of black land, and providing technical support for further improving the conservation tillage intelligence.

**Author Contributions:** Conceptualization, Z.M. and C.Z.; methodology, C.L., J.C., Z.M. and C.Z.; software, J.C., S.G., X.A. and Y.Y.; validation, J.C.; resources, Z.M. and Y.Y.; data curation, C.L. and J.C.; writing—original draft preparation, C.L., J.C. and Z.M.; writing—review and editing, C.L. and Z.M.; supervision, C.Z.; project administration, Y.Y., C.W. and H.L.; funding acquisition, Z.M. All authors have read and agreed to the published version of the manuscript.

**Funding:** This research was funded by the National Natural Science Foundation of China (Grant No. 31971800) and National Key Research and Development Plan of China (Grant No. 2020YFB1709602).

**Institutional Review Board Statement:** Not applicable.

**Informed Consent Statement:** Not applicable.

**Data Availability Statement:** All data are presented in this article in the form of figures or tables.

**Acknowledgments:** We would like to thank the "Intelligent Equipment Research Center, Beijing Academy of Agriculture and Forestry Sciences" and "National Engineering Research Center for Information Technology in Agriculture".

**Conflicts of Interest:** The authors declare no conflict of interest.

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
