# Peer review of "Development and Application of a Remote Monitoring System for Agricultural Machinery Operation in Conservation Tillage"

_agriculture, doi:10.3390/agriculture12091460_

Round 1

Reviewer 1 Report

It is not clear from the literature review whether such methods of tillage monitoring have been used before or whether such a method was proposed for the first time? If they have already been used, then it is worth noting them and reflecting what is the difference between the proposed method and the existing one.

The authors of the article refer mainly to the works of scientists from China, it is necessary to expand the review and present the developments of scientists from other countries.

At the beginning of the description of the methodology of the experiment, it is necessary to specify which straw is used (only in subparagraph 2.3 it is indicated that corn straw is used). In addition, it is necessary to indicate why corn straw is used and whether this method presented in the article can be applied to other crops, for example, wheat.

Can this system of determining the depth of tillage be used when using disc plows, or is it used only for ploughshares?

The conclusions should reflect the numerical values obtained in the experiment when analyzing the depth of tillage and the degree of straw coverage, such as the degree of reliability of the results from the application of this method, the average error, etc.

Also, in the conclusions, it is necessary to compare the obtained results with the results of research by other scientists (including those presented in the introduction when reviewing scientific papers) on the grounds of the accuracy and efficiency of the proposed methods.

To simplify the search for articles referenced by authors, it is necessary to add doi in the References section.

Author Response

Thank you for your approval of this article. I have carefully revised the article according to your questions and suggestions. The specific modifications are as follows:

Reviewer 2 Report

Improving the level of agricultural mechanization can improve the utilization rate of agricultural resources, labor productivity and land output rate, thus promoting high quality and efficient development of agricultural mechanization and ultimately ensuring sustainable agricultural development. This study tries to proposes a dual-attitude composite tillage depth measurement method based on the tractor body and the lower pull rod of the three-point hitch mechanism and an online measurement method for the straw coverage rate on the field surface. Although the topic is very interesting, it suffers from major limitations. There are some issues that should be addressed.

Specific comments:

1. The manuscript lacks a condensation of innovation. Writing ideas are not clear, unable to highlight the focus of research. Manuscripts lacks theoretical depth.

2. In LINE 163. Figure 3 Non-standard flowchart drawing

3. Authors used camera to capture images for straw coverage detection. However, in the actual working environment there are situations such as dust and camera obscuration, which the authors did not consider how to deal with. And the authors did not explain how to evaluate the straw coverage in the test results, and the accuracy of the test results needs to be clarified

4. In LINE 230,Equation 3 is incorrect.

5. Parts of the System main function modules in the manuscript are not required. The manuscript also does not elaborate on this section.

6. I did not find Discussion in this manuscript. The authors only list the experimental data but do not analyse them, nor do they analyse and discuss the errors that exist

Author Response

(The authors gave the same response as above.)

Reviewer 3 Report

This manuscript developed a system for monitoring the operating status (tillage depth and straw coverage rate) of plough pulled by a tractor. A dual-attitude tillage depth measurement method was proposed for tillage depth detection, and image processing method was used for straw coverage rate estimation. Test results validated the effectiveness of the monitoring system. Though the innovation of the work is kind of weak, this work is valuable for agricultural engineering to develop monitoring systems for intelligent agricultural machinery. 

The English language and organization is poor and must be improved significantly. Some detailed comments are as follows. 

1. Title: 'farm equipment telematics system', improper expression.

2. Line 24-25: What is the logic relationship between this sentence and the  one before?

3. Line 28: what does 'which' stand for here?

4. Line 39: promoting-->promote

5. Line 40: ensuring-->ensure

6. Line 40-45: Too long sentence, making it very confusing. Please reorganize. 

7. Line 45-51: Too long sentence, making it very confusing. Please reorganize. 

8. Line 46: 'agricultural machinery equipment', in the content of 'such as' I did not see anything about 'equipment'.

9. Line 49: what do you mean by 'information management control'?

10. Line 51: This?

11. Line 57-58: an essential means-->essential

12. Line 59: Yin Yanxin-->Yin. Usually only a given name is used for citation. Please revise through the whole manuscript. 

13. Line 59: Which reference in the reference list is cited here? Reference 8?

14. Line 59-74: Is there only one paper studying on tillage depth detection? More relevant papers should be reviewed and analyzed.

15. Line 76: ‘Straw returning…’ From here you began to introduce straw returning and straw coverage detection. It is suggested to start a new paragraph. Again, review of relevant research is too short and needs to be extended. 

16. Line 82:’The traditional straw coverage’, improper subject. You can traditional method for straw coverage detection.

17. Line 97: ‘most’, improper subject.

18. Line 101: ‘mechanized conservation tillage agricultural machinery’, improper expression.

19. Line 102-103: ‘method for subsoiling operation…rate’, should it be ‘method for …rate detection’?

20. Line 107: ‘has helped’, wrong tense.

21. Line 115: consistconsists. The English must be improved significantly!

22. Line 116: ‘surface scene image data’, improper expression.

23. Line 120: ‘to realize the …data’, grammar mistake.

24. Line 128: ‘calculates’, what is the subject?

25. Line 169-175: too long sentence and confusing. 

26. Line 187: ‘place’, what is the subject?

27. Figure 5: where is the angle gamma?

28. Line 230, 231, 235, 238, 240: wrong usage of semicolon.

29: Section 3: The review thinks this whole section is just about engineering. There is no scientific problems to be solved. And your experiment did not show any result about the remote monitoring system. So it is unnecessary to introduce this system in such a detailed manner.  You may just introduce the main modules briefly, like the first paragraph of Section 3.1. All the detailed introduction about this system can be removed. This is a scientific paper rather than a user manual.

30. Figure 10: (a) Testing site (b)Test site?

31. Line 367: iswas. All your experiments were conducted before writing this manuscript, so a past tense should be used. Please revise the tense through the whole manuscript.

32. For your tillage depth detection system, what is the sampling rate? Did you process the acquired signal? Please specify in the manuscript.

33. Figure 13: please add the true depth values into the figures for comparison. Did you test your system under different tillage depths to validate the robustness of your system? For tillage, a constant depth (like 30 cm) is preferable. However, due to different interferences, the tillage depth may vary in a range like from 20 cm to 30 cm. Therefore, a test of the system under different depths is necessary.

34. Conclusions: For an English paper, the conclusion is not organized in the style of ‘1,2,3’.

Author Response

(The authors gave the same response as above.)

Round 2

Reviewer 2 Report

The author made some changes to the comments made by the reviewers of the manuscript, but the reviewers still had some questions.

1. The author mentioned:“In this paper, the camera installed on the front of the tractor was set at 45° for image capture. Before each operation, the lens was wiped once. Shading components were installed on the front of the camera to ensure that the collected images were not affected by dust, bright light, and other factors.

In the process of working, the camera will not be affected by the outside environment?

2. In the revised manuscript, the authors still did not explain how to evaluate the straw coverage in the test results. How is straw coverage determined in field tests rather than in image processing?

Author Response

(The authors gave the same response as above.)

Reviewer 3 Report

Overall the authors have addressed all my questions and the English has been improved significantly. Some minor language errors can still be improved, such as:

1. Line 19:The operation monitoring->A operation monitoring

2. Line 20: The remote monitoring->A remote monitoring

3. ......

Author Response

(The authors gave the same response as above.)
